# Stem-Cell-Based Therapy: The Celestial Weapon against Neurological Disorders

**DOI:** 10.3390/cells11213476

**Published:** 2022-11-02

**Authors:** Mohamed A. Zayed, Samar Sultan, Hashem O. Alsaab, Shimaa Mohammad Yousof, Ghadeer I. Alrefaei, Nouf H. Alsubhi, Saleh Alkarim, Kholoud S. Al Ghamdi, Sali Abubaker Bagabir, Ankit Jana, Badrah S. Alghamdi, Hazem M. Atta, Ghulam Md Ashraf

**Affiliations:** 1Physiology Department, Faculty of Medicine in Rabigh, King Abdulaziz University, Jeddah 21589, Saudi Arabia; 2Physiology Department, Faculty of Medicine, Menoufia University, Menoufia 32511, Egypt; 3Medical Laboratory Technology Department, Faculty of Applied Medical Sciences, King Abdulaziz University, Jeddah 21589, Saudi Arabia; 4Regenerative Medicine Unit, King Fahd Medical Research Center, King Abdulaziz University, Jeddah 21589, Saudi Arabia; 5Department of Pharmaceutics and Pharmaceutical Technology, College of Pharmacy, Taif University, Taif 21944, Saudi Arabia; 6Medical Physiology Department, Faculty of Medicine, Suez Canal University, Ismailia 41522, Egypt; 7Department of Biology, College of Science, University of Jeddah, Jeddah 21589, Saudi Arabia; 8Department of Biological Sciences, College of Science & Arts, King Abdulaziz University, Rabigh 21911, Saudi Arabia; 9Embryonic and Cancer Stem Cell Research Group, King Fahad Medical Research Center, King Abdulaziz University, Jeddah 21589, Saudi Arabia; 10Biology Department, Faculty of Sciences, King Abdulaziz University, Jeddah 21589, Saudi Arabia; 11Embryonic Stem Cells Research Unit, Biology Department, Faculty of Sciences, King Abdulaziz University, Jeddah 21589, Saudi Arabia; 12Department of Physiology, College of Medicine, Imam Abdulrahman Bin Faisal University, Dammam 31441, Saudi Arabia; 13Genetic Unit, Department of Medical Laboratory Technology, Faculty of Applied Medical Sciences, Jazan University, Jazan 45142, Saudi Arabia; 14School of Biotechnology, Kalinga Institute of Industrial Technology (KIIT) Deemed to be University, Campus-11, Patia, Bhubaneswar 751024, Odisha, India; 15Department of Physiology, Faculty of Medicine, King Abdulaziz University, Jeddah 21589, Saudi Arabia; 16Pre-Clinical Research Unit, King Fahd Medical Research Center, King Abdulaziz University, Jeddah 21589, Saudi Arabia; 17Clinical Biochemistry Department, Faculty of Medicine in Rabigh, King Abdulaziz University, Jeddah 21589, Saudi Arabia; 18Medical Biochemistry and Molecular Biology Department, Faculty of Medicine, Cairo University, Cairo 11562, Egypt; 19Department of Medical Laboratory Sciences, College of Health Sciences, University of Sharjah, University City, Sharjah 27272, United Arab Emirates

**Keywords:** stem cells, therapy, neurodegenerative diseases, Parkinson’s disease, Huntington’s disease, Alzheimer’s disease, amyotrophic lateral sclerosis, neuropathic pain, brain ischemic stroke

## Abstract

Stem cells are a versatile source for cell therapy. Their use is particularly significant for the treatment of neurological disorders for which no definitive conventional medical treatment is available. Neurological disorders are of diverse etiology and pathogenesis. Alzheimer’s disease (AD) is caused by abnormal protein deposits, leading to progressive dementia. Parkinson’s disease (PD) is due to the specific degeneration of the dopaminergic neurons causing motor and sensory impairment. Huntington’s disease (HD) includes a transmittable gene mutation, and any treatment should involve gene modulation of the transplanted cells. Multiple sclerosis (MS) is an autoimmune disorder affecting multiple neurons sporadically but induces progressive neuronal dysfunction. Amyotrophic lateral sclerosis (ALS) impacts upper and lower motor neurons, leading to progressive muscle degeneration. This shows the need to try to tailor different types of cells to repair the specific defect characteristic of each disease. In recent years, several types of stem cells were used in different animal models, including transgenic animals of various neurologic disorders. Based on some of the successful animal studies, some clinical trials were designed and approved. Some studies were successful, others were terminated and, still, a few are ongoing. In this manuscript, we aim to review the current information on both the experimental and clinical trials of stem cell therapy in neurological disorders of various disease mechanisms. The different types of cells used, their mode of transplantation and the molecular and physiologic effects are discussed. Recommendations for future use and hopes are highlighted.

## 1. Introduction

In the 21st century, stem cells have gained tremendous importance in the fields of medical research and therapy. Stem cells are recognized as body cells that have unique characteristics, including the ability of self-renewal and differentiation into several type of body cells [1]. They can remain undifferentiated (totipotent) and are capable of differentiating into several mature cells [2]. Stem cells can be categorized depending on their sources: embryonic stem cells, fetal stem cells, adult stem cells and induced pluripotent stem cells [1].

In the medical field, stem cells have been authorized for use in bone marrow transplantation for the treatment of hematological malignancies and some inherited metabolic diseases. Recently, successful stem cell transplantation was reported to cure human immunodeficiency virus (HIV) infection [3]. Several experimental studies and/or clinical trials studied the use of different kinds of stem cells for the treatment of neurological disorders particularly the disorders lacking a definitive medical treatment. These include Alzheimer’s disease (AD), Parkinson’s disease (PD), Huntington’s Disease (HD), amyotrophic lateral sclerosis (ALS), multiple sclerosis (MS), temporal lobe epilepsy (TLE), neuropathic pain (NP), and brain ischemic stroke (BIS) [4,5].

The significance of stem cells is derived from their reported ability to replenish damaged cells and tissues as well as their anti-inflammatory and immune-modulatory properties. Stem cells can differentiate and replenish cells that have been weakened or destroyed, promoting neural tissue growth and development. Most of the neurological disorders are characterized by the widespread neuronal death and the extremely low regenerative potential of the brain. The treatment needs materials or cells that can cross the blood brain barrier (BBB). All these factors contribute to making stem cell therapy a viable option for treating chronic intractable neurologic diseases [6,7]

By combining additional drugs, the results of stem cell therapy may be improved [8]. Stem cell treatment, for example, in combination with erythropoietin, had synergetic effects on rat neurogenesis. To overcome the limitations of stem cell migration and inclusion in functional networks [9,10,11], nanoparticle distribution systems are investigated. Since they cross the BBB and enter the target brain areas without affecting the surroundings, these nanoparticles are beneficial for drug and cell systems. Another choice for delivering and preserving stem cells in the transplant site is hydrophilic polymer encapsulation, thereby providing mechanical assistance in supply processes and increasing the proliferation and differentiation in hydrogels [12]. In recent research, gene therapy and neural development factors have also been used to extend the retention of AD and PD transplanted stem cells [13].

Many recent trials have shown the value of peripheral stem cell treatments for acute stroke survivors, enabling minimally invasive cell therapy to become a practical alternative option. Additionally, there is consideration of the use of mesenchymal stem cells (MSCs) for supplying bioactive factors, such as brain-derived neurotrophic factors (BDNF), in treating neurologic disorders, such as HD [14,15,16].

This review article aims to review the current research on the use of stem cells both experimentally and in clinical trials for the treatment of selected neurological disorders. The molecular mechanisms of stem cell regenerative actions will be detailed. The successful and promising therapeutic modalities will be emphasized.

## 2. Selection of Transplant Recipients

Cell transplantation requires several sets of issues that must be considered in preclinical and clinical trials [17]. The first is whether the disease causes brain cell death or initiates a transition in cell interactions. The next is the probability of systemic transplantation, which happens only if the blood–brain barrier (BBB) is known to be open. Another aspect is whether pathology causes an inflammatory response in relation to the condition itself. In this case, the transplanted cells can play not only an alternative role, but also an anti-inflammatory role. All critical issues must be considered for nearly all cells used in clinical trials [17].

Given the above, it is not easy to choose the cells to be transplanted. Many various stem cell types play a possible therapeutic role in the treatment of neurological conditions. Cells must play either a substitutional or trophic role. There was a great focus on the substitution feature in earlier years of stem cell transplantation in neurological disorders. Many experimental studies in animal models showed that transplanted cells in the nerve tissue did not produce a physiological response [18,19]. If the neural dysfunction is still significant and the surgical approach takes place, the tissue’s reconstitution and hence the rebuilding of the injured neural pathways are very complex. Spinal cord damage is associated with the loss of motor neurons with long axons covered by the myelin sheath. In such circumstances, the transplanted cells must replenish neurons and glia; however, to exert their therapeutic activity, they must be capable of expanding their processes in the right direction. It is unlikely that this challenge will be completed with the current expertise, but a potential approach for transplant research in sophisticated tissues may be the fusion of transplant therapy and bioengineering (scaffold construction) [12].

No biomarkers or successful medicines were able to slow down disease development, despite the billions in dollars of clinical trials and considerable advances in studying neurodegenerative mechanisms. This makes stem cell therapy for the treatment of neurodegenerative diseases a beneficial approach to be tested [20]. The first aim of stem cell therapy involves determining distinct neuronal subtypes and the recapitulation of a network of neural diseases similar to those lost. The development of environmental reinforcement in support of host neurons by the production of neurotrophic and scavenging toxic factors and the construction of an auxiliary neural network across the affected areas is another approach to the treatment of neurodegenerative disorders [21]. Another strategic approach is the synthesis of neuroprotective growth factors in the sites of diseases (e.g., glial-derived neurotrophic factor (GDNF), brain-derived neurotrophic factor (BDNF), the insulin-like factor 1 growth factor (IGF-1), and VEGF).

One of the biggest concerns is immune rejection of transplanted fetal tissue or cells, which can trigger serious host responses [22]. Although the brain is considered immuno-privileged, few human leucocyte antigen cells remain matched to the haplotype, requiring immunosuppression in recipients to prevent cell-induced immune refusal. While there are a few exceptions, new technologies are necessary to increase donors’ and recipients’ compatibility and avoid further immune rejections [23,24].

## 3. Stem Cells in Alzheimer’s Disease (AD)

Alzheimer’s disease is a widespread chronic, pathologically marked neurodegenerative condition with ß-amyloid plaques and neurofibrillary tangles. Current alternatives to medication only relieve the symptoms without treating the illness, which is a significant problem that impacts the quality of life of patients and their care providers. Stem cell therapy may offer new opportunities for AD patients’ care. More and more research has shown that neural stem cells(NSCs) developed from embryonic stem cells (ESCs) were efficient as a treatment approach in AD models, showing changes both in vitro and in vivo [25,26]. Stem cells have the potential to differentiate from the brain extracellular matrix into neural cells, and they may restore neuroplasticity and neurogenesis via neurotrophic factors [27].

The stem cell approach to treating AD was first examined in animal models [28]. Neural stem cells from neonatal rat brains were used to establish new cholinergic neurons and increased learning and memory in rats with AD [29]. Neuron-like embryonic stem cells were used to restore AD-damaged rat brains [30]. Lately, the most used cells in Alzheimer’s disease research were embryonic stem cells (ESCs), mesenchymal stem cells (MSCs), brain-derived neural stem cells (NSCs), and induced pluripotent stem cells (iPSCs) [31].

The basal forebrain cholinergic neurons (BFCNs) are critically implicated in memory and learning disorders, such as AD [32]. The ESCs possess the pluripotency potential, which is double-bladed. Although that pluripotency is a great advantage for ESCs, it is a considerable disadvantage, as it may lead to the differentiation of the ESCs into several directions, which ultimately can lead to the formation of teratomas and tumors. Moreover, there is a tendency for eliciting disturbed immune reactions and rejection on transplanting ESCs [33,34]. Therefore, although the ESCs showed promising results in rat models of AD and improved memory performance, it has limited clinical applications. Two types of ESCs have been used in AD research: mouse ESCs (mESCs) and human ESCs (hESCs). Both produced the differentiation into BFCNs when transplanted in mice models with AD.

The immune rejection that occurs using allogeneic hESCs can be mitigated by transplanting iPSCs. Therefore, stimulating the differentiation of autologous hiPSCs could be promising in diminishing the chances of such immune rejection. Nevertheless, there are many concerns regarding the safety of using the iPSCs including the potential risk of oncogenesis and teratoma formation, the safety of the long-term use, the reprogramming efficiency, and immunogenic liability. The partial reprogramming and the unstable genes might elicit an immunological reaction with iPSCs. There is a need to develop new methods and protocols to avoid the expression of the tumorigenic genes when using iPSCs [34,35]. The iPSCs can differentiate into different cell types, including neurons. In AD research, iPSCs can be used, for example, to investigate the inflammatory reaction, to induce macrophages that can express a protease that degrades beta-amyloid called neprilysin and to reprogram the fibroblast and hence identify the phenotype of the AD [31,36,37].

Despite the ethical issues, the most widely used type of stem cells utilized in AD research is MSCs obtained from umbilical cord blood. This is attributed to the feasibility of obtaining umbilical cord blood after delivery [31,38]. Previous reports have shown that MSCs can improve the deficits in memory and learning in AD murine models. Boutajangout et al. reported that human umbilical cord mesenchymal stem cell (HUC-MSCs) xenografts improved cognitive decline and reduced the Amyloid burden in a mouse model of Alzheimer’s disease [39].

Many mechanisms have been suggested to be involved in this process, including decreased beta-amyloid plaques, a dramatic decrease in β-secretase 1 (BACE-1) levels, reduced hyperphosphorylation of tau, and the reversal of the inflammatory process in the microglia as well as the enhancement of anti-inflammatory cytokines [40]. Additionally, the immunomodulation and anti-inflammatory effects of MSCs have been reported to occur via enhancing the neuroprotection and depressing the proinflammatory cytokines. Moreover, bone marrow MSCs have been found to stimulate the formation of extracellular vesicles and microvesicles. These vesicles, in turn, target the amyloid-beta [41,42]. Additionally, the evolved plasticity and neurotrophic decline, decreased tau phosphorylation, and neuroinflammation, and tau down-regulation are promising targets for stem cells. Results showed the transgenic mice survived without any harmful effects and showed increased memory. This was confirmed in another study, where synaptogenesis increased the mental capacity in mice [43,44]. The successful preliminary animal studies showed positive findings. Researchers grafted human umbilical mesenchymal stem cells obtained from donor cords, into AD mice. An anti-inflammatory and immune-modulatory reaction was simultaneously induced in the mice by this study, and the presence of M2-like microglia enhanced synapsin and raised A β levels in the brain, thereby decreasing amyloid accumulation [45]. This led to a clinical trial in 2015 using human umbilical cord blood-derived mesenchymal stem cells (hUCB-MSCs) on nine patients with mild-to-to-moderate AD. The hUCB-MSCs were stereotactically inserted into the hippocampus. The method of administration of the stem cells was stable and feasible, with no consequences. Better studies with larger sample size and placebo monitoring are needed to advance the hypothesis. The administration of stem cell therapy was safe but needs to be further checked for its therapeutic efficacy on AD pathogenesis. [46]. The outcome of some studies on Alzheimer’s patients is still unknown such as (NTC01547689, NTC02672306, NTC02054208, and NTC02600130 from Clinicaltrials.gov). Nevertheless, they are all experiments constrained by the variation of neurons affected by AD.

The biotechnology company Nature Cell has begun a new phase II clinical trial using a stem cell medicine for AD (AstroStem) consisting of autologous adipose tissue stem cells administered intravenously into 60 AD patients (200 million cells/injection) (NCT03117738).

## 4. Stem Cells in Parkinson’s Disease (PD)

Parkinson’s disease is a degenerative disorder characterized by nigrostriatal dopaminergic neuronal loss. There are about 10 million patients globally [47]. Till now, there have been no medications that stop the dopaminergic neuronal degeneration. The PD hallmarks represent the cytoplasmic accumulation of α-synuclein and synthesis of Lewy bodies in dopaminergic neurons affecting several regions of the central nervous system [48,49]. It causes motor impairments, such as muscular rigidity, bradykinesia, static tremors and postural instability [50]. In addition, PD patients presented with other manifestations, such as sleep and behavioral disorders and abnormal GIT motility [51]. Because no actual treatment can halt neuronal degeneration, stem cell transplantation is a promising therapy that can restore dopamine (DA) neurotransmission and replace the lost neurons.

Early studies in PD animal models transplanted with mesencephalic cells formed neuronal protrusions with dopamine formation [52,53,54,55,56]. Additionally, human fetal ventral mesencephalic transplantation in PD patients in clinical studies declared the moderate improvement of PD symptoms [57,58,59,60,61]. However, the postmortem investigation of patients that transplanted fetal ventral mesenchephalic (fVM) tissue grafts showed the presence of Lewy bodies in the transplanted cells [62], suggesting that Lewy body pathology can spread from host to graft [63]. Additionally, graft-induced dyskinesia was a side effect of fVM transplantation. Indeed, most of successful results were achieved in PD cases below the age of 60, shown in Figure 1 [64].

Multiple technical problems appear before the use of fVM grafts in clinical practice. The first problem is the limited viability of grafted cells in the host striatum. Attempts were made to deal with this issue through the supply of several neurotrophic factors [65]. The second challenge is the minimal accessibility of human fetal cells, and the variations in protocols. Notably, immunosuppressant intake is required to avoid allograft-induced immune rejection. We need up to seven human fetal donor cells for each patient, and this creates actual ethical concerns [66,67].

To overcome these contests, the European consortium TRANSEURO, a multicenter for clinical trials, works to assess the feasibility and efficacy of human fetal cell transplantation in PD cases and provide more reliable results and more understanding of the potential therapeutic benefits [67].

Human embryonic stem cells (hESCs) are pluripotent stem cells located in the inner layer of early embryonic blastocysts [68], and they can differentiate into multiple cells via various differentiation protocols in vitro [69,70]. By using hESCs, there was a remarkable expression of dopaminergic neuronal markers together with improvement of motor defects [71,72]. However, the major challenge is difficult control embryonic cell maturation and cellular heterogeneity that result in poor results in clinical application [73,74]. Another major problem is the risk of tumor formation [75,76].

The stem cell research was revolutionized after the reprogramming of human fibroblasts to pluripotent cells [77,78,79]. iPSCs have the same characteristics as hESCs but have a relatively easier extraction process. The most important point is non-invasive tissue collection as it can be extracted from skin fibroblasts, mononuclear blood cells, and umbilical mesenchymal cells [80,81,82,83]. iPSCs could differentiate into patient-specific neurons in vitro [84,85,86]. Additionally, the autologous transplantation of stem cells is important to minimize tissue rejection [87]. The efficacy of DA neurons derived from iPSCs was like that of hESCs [88,89]. Animal studies showed the effective improvement of symptoms after iPSCs-derived neuron transplantation in PD animal models [90].

In a study on the PD monkey model, the transplantation of autologous iPSC-derived DA neurons exhibited an obvious improvement in motor functions without immunosuppression therapy [87]. MSCs were confirmed to affect the management of multiple diseases, including PD [91]. MSCs cause improved PD symptoms in PD mouse models. Some improvements depend on the secretion of several neurotrophic factors that minimize DA neuronal degeneration [92,93].

In 1987, Professor Madrazo headed a team that recognized neural grafting as a new method for substituting missing dopamine cells. Two young PD patients were autografted with adrenal medulla tissue into the brain, leading to improved PD symptoms, including tremors, rigidity, and akinesia. Cell-based neural transplant and treatment have since been regarded as potential treatments for PD as a suitable candidate for a focal degeneration disorder [94]. Pilot research was conducted two years later including 18 patients who confirmed the findings of Madrazo. However, the technique has been discontinued because of insufficient preclinical evidence and patients having mental disorders after surgery [95].

These groundbreaking experimental studies are underway in stem cell therapy for Parkinson’s disease and have brought a new and unprecedented revolution to clinical trials. Thus, therapeutic translation instructions and treatment protocols were determined for the patients. With this sort of technology, in 2018, a Chinese team hoped to implant neural precursor cells that were obtained from human embryos into people with Parkinson’s disease. The research studied how the dopamine-producing neurons can progress to maturity. The world’s first neural stem cell transplant was completed in Australia by neurologist Dr. Andrew Evans of the Royal Melbourne Hospital and his colleagues. A few Australian patients were administered artificially generated, parthenogenically derived neural stem cells in a phase I safety study (NCT02452723). Furthermore, the iPSCs production by Takahashi and Yamanaka and personalized neural progenitor cells from stem cells give a great source of patient-specific and disease-specific neurons, avoiding many of the problems with hESC lines (on both ethical and environmental fronts) [78,96]. In light of the recent progress in iPSCs, it was concluded that the most critical iPSCs priorities are studying the human disease mechanisms, and that is why iPSCs studies are scarce in clinical trials [97].

In 2017, a team of Japanese researchers found a safe and efficient method to create neurons that can make dopamine from patient-derived iPSCs cells. By the process of direct midbrain dopaminergic progenitor grafting into the monkeys, they treated them with 1-methyl-4-phenyl-1,2, 3,6-tetrahydropyridine (MPTP), which ablates nigral dopaminergic neurons to watch their behavior for two years to see how it will decrease their symptoms [89]. The researchers’ hopes were dramatically raised, as the monkeys recovered with stable motor functions and with no cancer evidence of tumor growth in both PD families. Therefore, these tests proved to these researchers that the technique could be used on people.

## 5. Stem Cells in Huntington’s Disease (HD)

Huntington’s disease is a degenerative autosomal dominant disorder characterized by GABAergic degeneration of medium spiny neurons (MSNs) in striatum, cerebral cortex, thalamus, and hypothalamus [98]. This degeneration leads to progressive deterioration of motor and cognitive functions [99]. HD represents multiple repeats of the trinucleotide CAG in the HD gene, causing increased expression of the Huntington (HTT) protein, which reduces brain-derived neurotrophic factor (BDNF) and causes medium spiny neurons (MSNs) [100] protein processing abnormalities [101] and improper mitochondrial function [102]. Despite the detection of the causative mutation more than 20 years ago, no effective treatment can halt HD consequences. Cell-based therapy is an attractive approach to treat HD disease. The principal scope is to restore the degenerated neurons and supply neurotropic support to avoid more deterioration; see Figure 2.

Traditionally, embryonic, and fetal striatal tissue grafts were utilized in animal models of HD [103,104,105]. These studies declared that the transplanted fetal tissue successfully differentiated into striatal tissue could improve cognitive and motor functions [103,105,106,107,108,109,110,111,112]. The usage of fetal stem cell therapy in HD is faced with many obstacles, concerns, and ethical, technical, and safety issues. Some studies showed some side effects of fetal cell transplantation such as graft cell overgrowth, and graft tissue rejection [113,114]. While immunosuppressive treatments can limit these reactions, prolonged immunosuppression raises additional safety concerns [115].

Neural stem cells were taken from a fetal brain. This procedure still needs the usage of aborted fetuses, but the ability to generate immortalized lines of hNSCs produces homogenous cell groups with lower tissue requirement. Ryu et al. stated that hNSCs transplantation to the HD rat model enhanced motor functions when the hNSCs were grafted before HD lesion, while cell transplantation after induction of HD lesion did not improve motor function [116]. The transplanted hNSCs displayed the endogenous secretion of BDNF, indicating that the motor enhancement may be caused partially by neurotrophic secretion by the transplanted cells. Another study declared significant improvement in the motor function and reduction of striatal atrophy after hNSCs transplantation in HD model with the existence or absence of the ciliary-derived neurotropic factor supply before transplantation, representing the role of transplanted cells in the production of further neurotrophic support [117]. Other studies demonstrated that hNSCs migrated into the striatum and enhanced motor functions when they were injected to the ventricles or intravenously in HD rats, giving attention to use this less invasive method in the clinical studies of HD patients [118,119].

Human pluripotent stem cells took a remarkable consideration as a cell-based therapy for neurodegenerative disorders. Early preclinical studies declared that hESC-derived neuronal stem cells enhanced motor functions in the HD model. However, the transplanted stem cells did not exhibit differentiation into MSNs [120,121]. To overcome this challenge, groups of hESCs differentiated into lateral ganglionic eminence progenitor cells or into striatal precursor cells were used to enhance cell MSNs’ specified differentiation [122,123,124,125,126]. While several studies declared proper integration of the transplanted cells and improvement of motor functions [122,123], other studies did not achieve the same results [124,125]. These discrepancies may be due to variability in the cell maturation and the transplanted cell numbers between studies [123,125,127].

Induced pluripotent stem cells are another a source to be used in cell replacement therapy of HD [128,129]. iPSCs have the advantage that they are collected from the patient, so they allow for autologous transplantation, which avoids immune rejection and the need for immunosuppressive therapy after transplantation [130,131]. In a study, iNSCs derived from fibroblasts were transplanted into the striatum, differentiated to neurons that expressed MSNs markers in mice [132]. Moreover, the transplanted cells exhibited prolonged survival for more than 6 months. However, the nature of HD as a genetic disorder restricts the usage of these autologous cells for cell therapy. Therefore, the genetic modification of the HD mutation becomes an important target before autologous cell therapy. There was a study that corrected the HD mutation in iPSCs before transplantation into HD mouse model [129]. Although this study did not show motor improvement, the transplanted cells were differentiated into neurons with the expression of MSNs markers.

Another promising study discussed the combination of stem cell and gene therapy in a transgenic model of HD in mice by using the neural progenitor cells of a rhesus monkey. Genetically adjusted NPCs showed a significant prolonged lifespan and enhancement of motor functions [133]. While there was no evidence of tumorigenesis after hNSC transplantation therapy in rats, the short lifespan of rat and mice models does not produce adequate observation of prolonged side effects [115]. Thus, it is important that these procedures are tried on other animals with a comparable lifespan to humans before using these remedies in HD patients.

An alternative method to obtain neural progenitor cells is by obtaining NSCs directly from neurogenic areas of the adult brain [134,135]. An initial study utilized NSCs extracted from the subventricular zone for transplantation into the striatum of the HD rat model. This procedure led to a remarkable enhancement of motor functions [135]. About 15% of transplanted cells expressed MSNs markers. Moreover, the handling of NSCs with lithium chloride before transplantation elevated the percentage of MSN neurons to 34% [134].

## 6. Stem Cells in Amyotrophic Lateral Sclerosis (ALS)

Amyotrophic lateral sclerosis is a degenerative disorder that affects motor neurons leading to atrophy and weakness of skeletal muscles [136]. Alteration in executive function occurs in about 50% of patients, and about 15% of patients represent frontotemporal dementia [137]. There is no real therapy for ALS. So, stem cells are an optimistic approach that may restore degenerated neurons. The transplanted stem cells and derived motor neurons require developing long axons, and synapse with endogenous neurons and muscles [138,139]. ALS is a lethal neurodegenerative disease, which affects both the upper and lower motor neurons. Only about 10 to 15% of the cases are familial, whilst the sporadic cases represent most cases. The mechanisms underlying ALS are still elusive and many previous reports documented a variety of proposed mechanisms. Amongst these mechanisms are dysfunction of mitochondria, oxidative stress, and glutamate toxicity. Therefore, the development of a therapeutic protocol that affects multiple suggested mechanisms could lead to a more favorable outcome [140,141].

There is only one documented drug for managing ALS, riluzole, in Europe, and there have been no other newly registered drugs since 1994. In Japan and the USA, a newly emerging drug, edaravone, has been registered [142]. Stem cell therapy in ALS disease is based on “neighborhood theory”, where the transplanted cells secrete neuroprotective substances that limit the process of neurodegeneration. Transplanted stem cells, also, differentiate into astrocytes and microglia, or into other neurons, which connect with the affected motor neurons [140].

Most of the preclinical studies conducted on ALS models expressed superoxide dismutase 1 mutation (SOD1) [143]. SOD1 is the most prevalent mutation of ALS [144]. In an early study, the intravenous injection of umbilical cord blood delayed motor symptoms expression and prolonged survival in the ALS mice model [145,146]. Histological investigation showed that the injected cells infiltrated the regions of cerebral and spinal cord degeneration with the expression of neural biomarkers [147]. In addition, inflammatory cytokines were reduced in the degenerated areas [148]. In another study, grafted neurons from ESCs were differentiated into motor neurons and transplanted to the spinal cord of ALS rats [149]. Grafted rats showed improved motor functions but were exposed to paralysis later. The initial improvement of motor functions indicated that the engrafted motor neurons exhibit some benefits, probably by secretion of neurotropic factors.

Hematopoietic stem cells [150] and glial restricted precursors [151] improved motor functions in ALS rats. In addition, bone marrow transplantation from wild rats to ALS mice models delayed the progress of disease [152], while intravenous injection of bone marrow-extracted stem cells to ALS mice prolonged lifespan [153]. The transplantation of olfactory ensheathing cells delayed the appearance of symptoms; these cells were differentiated into oligodendrocytes and astrocytes [154], indicating that various cell types can alleviate ALS progress. The injection of mesenchymal stem cells to skeletal muscles of ALS rats produced a neuroprotective action via secretion of the glial-cell-derived neurotrophic factor [155] or vascular endothelial growth factor [156];

Another source of stem cells is adipose-derived MSCs. Preclinical studies showed the potential effect of adipose-derived MSCs in ALS mice model [157]. When adipose-derived MSCs were injected intravenously, the motor deterioration was slowed. Histological investigation of spinal cord detected a greater number of motor neurons in adipose-derived MSCs injected mice [158]. Additionally, the transplantation of NPCs into the spinal cord improved the disorder phenotypes in ALS mice, implicating the neuroprotective role of neurotrophins [159]. The transplantation of NPCs programmed to secrete GDNF, IGF-1, VEGF and brain-derived neurotrophic factor showed a variable level of success [160,161,162].

iPSCs are another source of autologous cells that have a great promise for treating ALS [163]. The utilization of human fibroblast cells programmed into iPSCs and differentiated to NPCs showed that the transplanted cells integrated into the spinal cord [164]. Another study compared the use of single versus multiple bone marrow-derived hMSCs injections into the cerebrospinal fluid in the SOD1 transgenic mice model of ALS. The researchers found no effect of the single hMSCs transplantation, whilst the multiple injections of hMSCs could have therapeutic potential against ALS [165]. An interesting study where the researchers administered adipose-derived MSCs (ADMSCs) into (SOD1)-mutant transgenic mice early in the clinical course of the disease documented postponed motor disturbances as assessed by electrophysiological methods and clinically. Additionally, they investigated the underlying mechanisms of this improvement. They suggested that the neuroprotective effect of the ADMSCs either directly or indirectly is behind the long-lasting powerful positive effect on motor function. The direct effect was obtained via the upregulation of the glial-derived growth factor, which is produced by the ADMSCs, while the indirect effect was suggested to be via the modulation of the local glial cells secretomes to be neuroprotective [157].

Despite that, stem-cell-based therapy in treating ALS is still in the beginning stages. Some clinical trials have been conducted in many countries around the world [140]. For example, a study that introduced Wharton’s jelly derived mesenchymal stem cells for patients with ALS reported the safety and effectiveness of this therapeutic approach. The study reported that the clinical and demographic characteristics, excluding sex, have no significant implications on the outcome. The study reported that the predictors for an effective outcome are female sex and good response to the first dose of therapy [141]. Another Chinese study introduced olfactory enhancing stem cells (OESCs) intracranially and documented preliminary outcomes about controlling or reversing the physiological derangements in ALS patients [166]. Nevertheless, other studies that utilized similar protocols showed contradicting results [140]. Therefore, the introduction of stem cell-based therapy for patients with ALS is still questionable and in need of further pre-clinical and clinical research before accepting it as a line of management of ALS.

## 7. Stem Cells in Multiple Sclerosis (MS)

Multiple sclerosis is an autoimmune disorder characterized by neuronal loss and demyelination [167]. Disease-modifying therapies for MS can reduce the severity of the disease. There is no successful therapy to stop the progress of the disorder and repair the present neural damage [168,169]. In the last two decades, stem cell therapy has been considered a potentially attractive method for MS.

Immunoablation therapy followed by autologous hematopoietic stem cells (aHSCs) transplantation was studied following the findings of the improvement of autoimmune phenoyptes in patients that undergo bone marrow transplantation for hematological malignancies [170,171]. In studies conducted on rats with experimental autoimmune encephalomyelitis (EAE), immunoablation followed by aHSCs promoted remission and avoided relapses [172,173].

The rationale of this approach is to remove the present immune system by using high-dose immunosuppressive therapy (HDIT), and develop a new intact immune system following aHSCs transplantation [174]. Earlier studies were directed to patients with advanced disease and higher disability [170,175,176]. Recent studies concentrated on patients with relapsing–remitting MS (RRMS) with poor prognosis, showing that aHSCs transplantation is most successful in these patients [177,178,179,180]. A study carried out on patients with RRMS reported 5-year event-free survival of 69% of patients [181].

MSCs were reported as a hopeful cell therapy for neurodegenerative disorders [182,183]. These potential results may be caused by paracrine signaling and the integration of MSCs in the damaged tissues [184]. When BM-MSCs were administered to EAE mice model, there was a diminution of the symptom’s severity, decrease in the infiltration of immune cells, and decrease in demyelination and axonal injury [184,185,186,187,188,189]. These results were shown when MSCs were injected intraventricularly, intravenously and intraperitoneally [187,190,191]. MSCs might influence differentiation of neuronal stem cell and induce axonal remyelination [192]. Histopathological analysis of EAE mice infused with MSCs reported the repairing of white matter, and enhanced axonal integration in diseased tissues [185].

Neuronal stem cells (NSCs) had a potential therapeutic effect as they migrated to the demyelinating tissues and differentiated into oligodendrocytes [193]. A major challenge is the fewer NSCs available to be used as an endogenous cell therapy. IVT-transplanted NPCs to EAE mice migrated into the injured white matter and differentiated to oligodendrocytes. However, the direct effect on myelin remyelination is still unclear [194].

Human embryonic stem cells (hESCs) differentiate to neural cell lineage, and this has attracted consideration as a potential approach to manage MS [195]. The transplantation of hESC-derived neural progenitors caused an improvement of symptoms in EAE mice. Histological analysis declared the integration of the transplanted neural progenitors in brain tissues. The improvement might be due to neuroprotective mechanisms [196].

Studies declared that oligodendrocyte precursor-derived induced pluripotent stem cells (iPSC) ameliorated the features of EAE [197]. This therapeutic effect was mostly due to the neuroprotective influence rather than remyelination. However, recent studies suggest the potential for malignancy. So, more research studies should be performed to overcome this challenge [198].

Multiple sclerosis typically presents with relapsing-remitting illness (RRMS), which is discrete and self-limited. However, after the remission of clinical symptoms, the damage that occurred in the brain does not resolve. Usually, it takes about 10–20 years for the RRMS to turn into secondary progressive MS (SPMS). Three major factors interplay to achieve the best efficacy of aHSCs transplantation in patients with MS. These factors are the selection of the patient, choosing the transplant regimen, and the experience of the working team [199]. In this section, we will focus on the patient section.

Previous reports documented that autologous stem cell transplantation (ASCT) began as pilot clinical began trials (Phase I/II) in Greece in 1995 for MS patients; in 2010, there were reports for 400 patients around the globe treated with ASCT [179]. Accumulating pieces of evidence point to the increased efficacy of the aHSCT in patients with RRMS when compared to patients with primary progressive MS (PPMS) or secondary progressive MS [177,200].

aHSCT should be introduced when there is an expected severe case of MS and should be done so early in the course of the disease, before irreversible neural damage occurs [1]. A variety of demographic and disease-related characters influence the outcome of the therapy by aHSCT. It was found that the patients with young age, patients with active inflammatory disease, patients with low scores of Kurtzke’s Expanded Disability Status Scale (EDSS) and patients without other co-morbidities show the better outcome of therapy by aHSCT [177,200,201]. The previously mentioned factors are interlinked to each other. For instance, when the patient has a shorter disease duration, this means that he/she is in the RRMS phase, which in turn points to better EDSS scores. It was found that patients with the active inflammatory stage of MS have better outcomes compared to patients with indolent inflammation. This is attributed to the ability of aHSCT to rapidly cease the inflammatory cells; see Figure 3 [199,202].

The gadolinium enhancement leads to the breaking down of the blood–brain barrier and hence allows the conditioned regimen to penetrate the CNS and, therefore, enhance the process of eliminating the autoreactive immune cells. The European Group for Blood and Marrow Transplantation (EBMT) published recommended guidelines for choosing patients who are eligible for aHSCT in 2012. They recommended aHSCT for patients who have RRMS and can ambulate independently, yet they suffered from two clinical relapses associated with MRI evidence of concurrent illness during the previous year in spite of the conventional disease-modifying therapies. Nevertheless, the individuals suffering from the inability to ambulate due to rapid progression of disability or sometimes those who have clear clinical or MRI evidence of progressive disease might be eligible for aHSCT. However, the outcome will be less favorable. Currently, there is a suggestion to use aHSCT in patients younger than 45 years old and with a disease course of less than 10 years [203,204,205]. Collectively, it is apparent that aHSCT is beneficial in treating MS during the different stages of the disease, however, for achieving a favorable outcome, the patient should be selected according to the recommendations of EBMT.

## 8. Stem Cells in Temporal Lobe Epilepsy (TLE)

Epilepsy is a neurological disorder which affects about 70 million patients all over the world [206]. TLE is the most prevalent type of epilepsy characterized by repeated seizures created in the structures of temporal lobe such as amygdala and hippocampus [207]. The loss of GABA-ergic neurons and synaptic changes were noticed in the hippocampus in patients and animal models with TLE [208], resulting in overall increased neuronal excitatory tone [209]. The current therapy of TLE mainly depends on anticonvulsant therapy or surgical interventions. About 1/3 of epileptic patients exhibit poor outcome when anticonvulsant therapy is used alone [210]. For refractory TLE, surgical intervention with temporal lobectomy is performed. However, the surgery results in reduced cognitive functions. So, there is an urgent requirement to find out a new therapeutic way for treatment of TLE. In the past years, different stem types of cells were used in preclinical studies to treat epilepsy in animal models.

Studies revealed that transplantation of ESCs could replace the damaged neurons and produce inhibitory mediators such as GABA which reduce neuronal excitability in epilepsy [211]. Another study reported that the transplanted ESCs into the hippocampi of status epilepticus mice model were differentiated to mature neurons [212].

As GABA has a role in reduction of excitability thus inhibition of seizures [211], transplantation of fetal GABA-ergic cells to the cerebral tissues had been studied in several epilepsy models. They hypothesized that elevation of GABA neurotransmitter level could inhibit seizure exposure [213,214]. The transplantation of fetal neural cells in epileptic rats minimized the severity of convulsive seizures, replaced degenerated neurons and repaired the damaged neural circuits [215].

Recent studies focused on fetal medial ganglionic eminence (MGE) cells for the treatment of TLE because MGE is the source of most striatal and hippocampal neurons [216]. The transplantation of fetal MGE precursor cells led to migration and synaptic formation in pyramidal neurons in the brain [217], and reduction in seizures number in mice [215,218]. The intravenous injection of fetal neural stem cells provided a remarkable inhibition in seizure rate and severity [219]. In addition, hippocampal precursor cells transplantation enhanced memory and learning defects in epilepticus status [220].

MSCs exhibited a neuroprotective property by releasing several neurotrophic factors and immunomodulation [221] which could inhibit seizure activity [222]. In a study, bone marrow MSCs transplantation decreased the chemical and histological alterations, restored neurotransmitters normal level, and reduced apoptotic and inflammatory marker levels in an epileptic rat model [223]. Another study revealed that adipose-extracted mesenchymal cell transplantation enhanced the release of neurotrophic markers and reduced seizure activity in an epileptic rat model [224].

## 9. Stem Cells in Neuropathic Pain (NP)

Neuropathic pain is a frequent complaint and usually occurs in diabetic patients with female predominance. Unfortunately, the current pharmacological therapies for neuropathic pain are mainly symptomatic, not fully effective, and have many side effects. Amongst these drugs are pregabalin (known commercially as Lyrica), tricyclic antidepressants, and opioids [225,226,227]. The administration of a combination of the available drugs may give a better outcome. However, the side effects remain to occur. Therefore, shifting toward stem cell therapy for neuropathic pain could provide a good chance for a better cure by affecting the underlying mechanisms of neuropathic pain instead of solely treating the symptoms; see Figure 4 and Figure 5 [225].

Among the crucial mechanisms of NP is the central sensitization, in which there is an enhancement in neuronal excitability. After nerve injury, there is an enhanced release of the excitatory neurotransmitter glutamate and upregulation of the N-methyl-d-aspartate (NMDA) receptors in the spinal cord. Studies showed that intravenous injection of bone marrow mesenchymal stem cells (BMSCs) inhibited the expression of NMDA receptors and hence prevented glutamate excitotoxicity [31,228].

One of the important mechanisms that lead to pain sensation in neuropathy is the glial cells (astrocytes, oligodendrocytes, and microglia) activation. Within 24 h post-injury, the microglia become activated and continue for about 12 weeks. This is followed by the release of cytokines from the microglia and astrocytes, along with the upregulation of the glutamate and glucocorticoids. This in turn can produce spinal cord excitation and therefore contribute to hypersensitivity and pain sensation. BMSCs were found to lower the microglial activity when injected intrathecally in case of non-compressive disc herniation. Additionally, it can lower the release of the inflammatory cytokines by the spinal cord-activated microglia and therefore affect behavioral hypersensitivity in nerve root pain [31,229,230]. Other studies found that intravenously administered ADSCs can decrease the expression of the astrocyte marker, GFAP.

Furthermore, amongst the mechanisms that modulate the effect of transplanted stem cells in NP are depressed apoptosis, autophagy, and promoting nerve recovery. It has been found that the expression of the apoptotic marker, TUNNEL, and LC3B-II and Beclin1 in the spinal cord are depressed in a burn fat model of neuropathic pain upon subcutaneous transplantation of adipose stem cells [231].

In addition to the previously mentioned central mechanisms that modulate the action of stem cells in neuropathic pain, there are many peripheral mechanisms, including powerful immunosuppressive and anti-inflammatory effects, enhanced expression of glial-derived neurotrophic factor (GDNF), enhancement of neurogenesis, neuronal growth, and myelin formation [231,232].

## 10. Stem Cells in Brain Ischemic Stroke (BIS)

It is generally stated that the transplantation of neural stem cells is an alternative therapy to replace damaged or dead neural cells or improve the self-repair system after brain ischemic stroke (BIS) [235]. In 2010, the first transplantation of a neural stem cell on clinical patients—phase I—with BIS was performed [236]. It was reported that after transplantation of the exogenous neural stem cells, the patients’ neurological function was improved without any unfavorable reaction. The number of clinical studies were few; however, there is a large number of preclinical animal studies [235,237,238,239]. These studies stated that neural stem cell transplantation therapy could enhance the functional outcomes as well as the histological outcomes where the infraction volume is considerably decreased. It has been reported that, essentially, the transplantation is focusing in two directions; the first strategy is to replace the dead cells, and the second strategy is to improve the self-repair system by mediating the neural network reconstruction. Moreover, it has been found that the degree of progress in function of BIS animal model has an effect on the injection time of neural stem cells and the source of stem cells [239].

Table 1 and Table 2 summarize the stem cells application in various neurological disorders.

## 11. Strategies to Enhance Cell Survival after Transplantation: Hypoxic Preconditioning and Genetic Modification

Recently, stem cell research included several creative strategies for treating neural disorders. These strategies aimed to resolve the problems of cell survival, differentiation, and engraftment rate after stem cell transplantation. They include preconditioning with different methods, such as hypoxia, drugs, or growth factors. They also include genetic modifications, upregulating the expression of specific growth factors, pro-survival, or anti-apoptotic genes of the stem cells before transplantation [249,250].

It has been reported that preconditioned neural stem cells with hypoxia before transplantation produced a significant increase in the rate of survival, proliferation, and differentiation as well as decrease in the number of apoptotic cells [251,252]. In addition, it also showed a significant increase in the biological function of transplanted cells [253]. The oxygen level of 1.5 to 5.3% was found to be efficient to increase proliferation of neural stem cells and differentiation into functional neuron cells [254]. The postulated mechanism is the activation of hypoxia-inducible factor1 (HIF1) transcriptional complex [255,256]. It comprises alpha (HIF1-α) and beta (HIF1-β) subunits. In mammals, it is the master regulator that facilitates important homeostatic responses to low oxygen levels. In the presence of oxygen, the HIF1-α is downregulated through hydroxylation by prolyl-4-hydroxylase and factor-inhibiting HIF enzymes [255,257]. These two hydroxylases inhibit the activation of the HIF system. Meanwhile, under hypoxic condition, HIF1-α will translocate to the nucleus and form dimerization with HIF1-β subunits. Then, the activated HIF system is attached to the hypoxia response elements, leading to activation of the expression of several essential genes, such as erythropoietin (EPO), vascular endothelial growth factor (VEGF) and glucose transporter 1 (GLUT-1) [258]. Those genes have a fundamental role in improving neural stem cell survival and treating ischemic stroke disorders.

Another way to improve cell survival and therapeutic outcomes after transplantation is to genetically modify the stem cell. The neural stem cell is genetically modified prior transplantation to overexpress neurotrophic support genes, such as brain-derived neurotrophic factor (BDNF), glial cell-derived neurotrophic factor (GDNF), VEGF, EPO and nerve growth factor (NGF). These produce enhanced survival rates and improved proliferation and neural differentiation [259,260,261,262]. In addition, they show immunomodulatory action and increase the expression of antioxidant and survival genes such as Bcl-2 [259,263]. Gretchen et al. (2018) used a genetically modified human cortical-derived neural stem cell to overproduce GDNF before transplantation into a rat cortex. These cells were able to differentiate into astrocytes producing GDNF [264]. Hypoxic preconditioning and genetically modified neural stem cell transplantation characterize a therapeutic strategy for neural disorders.

## 12. Conclusions and Future Perspectives

Previous studies indicated that stem cell therapy for neurological diseases may be successful in experimental disease models and in certain limited clinical trials. The selection of the most suitable cell type that suits a particular disease is a challenge. This is due to the diverse neuropathological effects of these diseases. Despite the drawbacks and complexities of stem cell treatment, the future of neurodegenerative disorders is still an exciting approach. It remains impractical and far to use stem cells in neurodegenerative diseases to substitute missing neurons and incorporate them into the original neural circuitry [265]. However, it is more practical and feasible in the short-term to use stem cells to provide therapeutic factors and inhibit disease progression. More information on the correct delivery strategy and immunosuppression, graft survival, and effectiveness will be provided by the continuing and prospective clinical trials. The development of new cellular origins and the further development of successful combination approaches to treating neuropathic disorders was possible through best practices in stem cell therapies. The use of ESCs and iPSCs, though pluripotent, may be complicated by teratomas and /or cancer. MSCs, whether derived from bone marrow or adipocytes, possess immense plasticity, but their mode of action and duration of survival in the host is questionable. Human umbilical cord stem cells provided some improvement in cognitive function in AD. The grafting of fetal and/or adrenal medullary tissues in PD had positive impact on the disease symptoms, but the patients suffered from neuropsychiatric complications. Additionally, the use of fetal tissue from aborted fetuses creates ethical concerns in different disciplines. In addition, long-term immunosuppression is needed after tissue transplantation. Combining the transplanted cells with certain drugs, such as erythropoietin, proved to be useful in experimental neurogenesis. Nanoparticle distribution systems cross the BBB and enter the target brain areas without affecting the surroundings; these nanoparticles are beneficial for facilitating the delivery of drugs and cell systems. Stem cells could be encapsulated in hydrophilic polymers, thus helping delivery and preservation in the transplant site [12]. Gene therapy and neural development factors have also been used to extend the retention of AD and PD transplanted stem cells [13]. Cell preconditioning with growth factors, exposure to hypoxia, and genetic modification proved to be useful at the experimental level. The idea is to provide a better fate and function to the transplanted cells. The combination and preconditioning therapies are gaining momentum and it seems that they will dominate the future work in this field.

## Figures and Tables

**Figure 1 cells-11-03476-f001:**
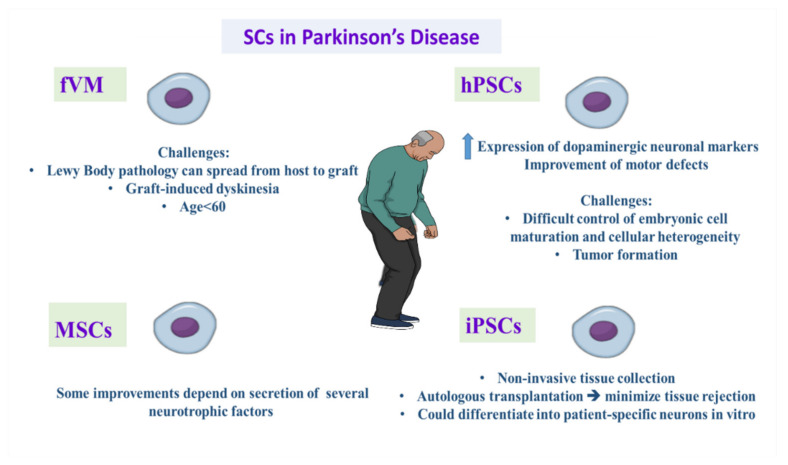
Demonstrating the promising role and challenges of different types of stem cell therapy in treating Parkinson’s disease. iPSCs: induced pluripotent stem cells, hPSCs: human pluripotent stem cells, fVM: fetal ventral mesencephalic tissue, MSCs: mesenchymal stem cells.

**Figure 2 cells-11-03476-f002:**
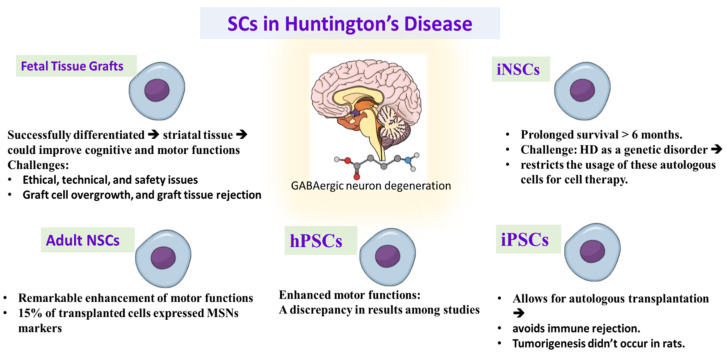
Demonstrating the promising role and challenges of different types of stem cell therapy in treating Huntington’s disease. iNSC: induced neuronal stem cells, iPSCs: induced pluripotent stem cells, hPSCs: human pluripotent stem cells, NSCs: neuronal stem cells, MSNs: striatal medium spiny neurons.

**Figure 3 cells-11-03476-f003:**
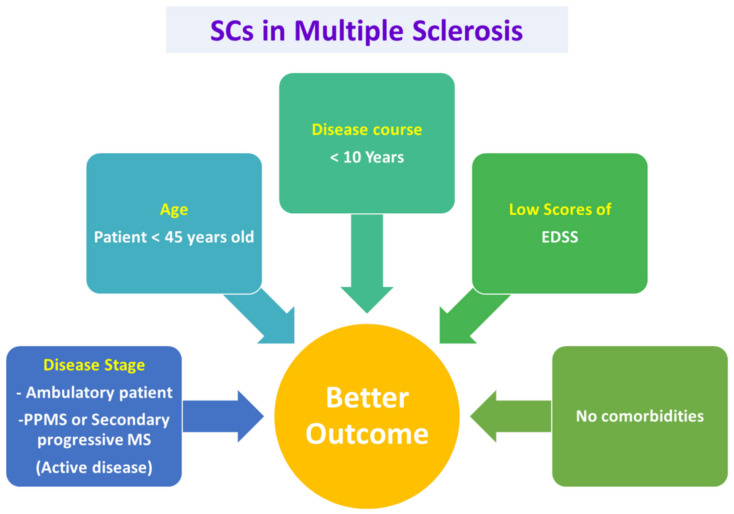
Selection criteria for patients of MS to be treated with aHSCs for a better outcome. Based on the EBMT and previous reports, Patients with early active disease stages, younger than 45 years old, with disease course lesser than 10 years and low scores of EDSS and no comorbidities are found to have better outcome on transplantation of autologous stem cells. EDSS: Kurtzke’s Expanded Disability Status Scale, EBMT: European Group for Blood and Marrow Transplantation.

**Figure 4 cells-11-03476-f004:**
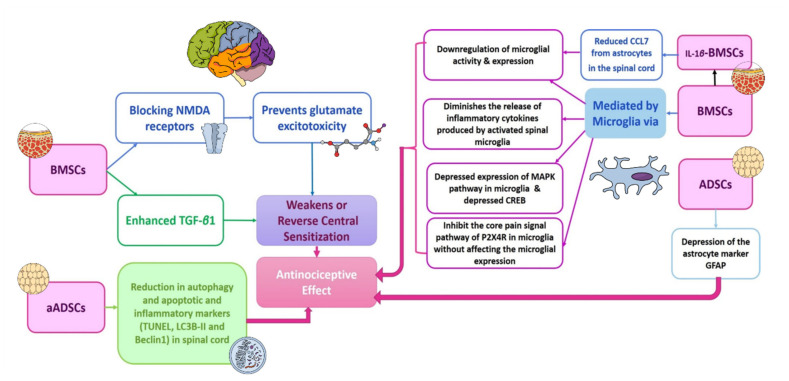
Proposed mechanisms for central NP alleviation via different types of stem cell therapy. Based on previous research, there are three main mechanisms for the antinociceptive effects of stem cell therapy. These mechanisms encompass the modulation of the glial cells (microglia and astrocytes) functions, weakening or reversal of central sensitization and reduction in autophagy, apoptosis and inflammatory markers in the spinal cord. Each of these main mechanisms are mediated via different ways according to the type of stem cells investigated. It is obvious that BMSCs act via different mechanisms to induce the analgesic effect. Additionally, conditioning of the BMSCs via IL-1β adds to these mechanisms. The ADSCs act mainly via depressing the astrocyte markers. Moreover, the transplantation of autologous ADSCs reduces the autophagy, apoptotic and inflammatory markers in the spinal cord. ADSCs: adipose-derived stem cells; aADSCs: autologous adipose-derived stem cells; BMSCs: bone-marrow-derived stem cells; CCL7: chemokine (C-C Motif) ligand 7; CREB: cAMP response element-binding protein; GFAP: autoimmune glial fibrillary acid protein; IL-1β BMSCs: IL-1β conditioned bone-marrow-derived stem cells; LC3-BII: autophagy marker light chain 3; MAPK: mitogen-activated protein kinase; NMDA: N-methyl-D-aspartate receptor; P2X4R: purinergic 2X4 receptor; TGF- β1: transforming growth factor beta 1; TUNEL: terminal deoxynucleotidyl transferase-mediated dUTP nick-end labeling.

**Figure 5 cells-11-03476-f005:**
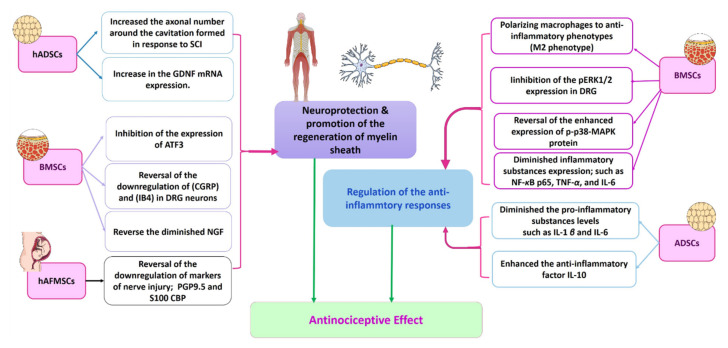
Proposed mechanisms involved in peripheral NP alleviation by stem cell therapy. Based on previous research, it is clear that there are two main mechanisms by which the different types of stem cells counteract hyperalgesia. These mechanisms are regulation of the anti-inflammatory responses and neuroprotection and promotion of the regeneration of myelin sheath. BMSCs act via the two mechanisms through a variety of pathways that varied according to the methodologies of the previous research. The ADSCs seem to regulate the anti-inflammatory responses, whilst the hADSCs work through the neuroprotective mechanism. The hAFMSCs act mainly via neuroprotection [31,233,234]. ADSCs: adipose-derived stem cells; ATF3: activating transcription factor 3; BMSCs: bone marrow derived stem cells; CGRP: calcitonin gene-related peptide CREB: cAMP response element-binding protein; DRG: dorsal root ganglion; GDNF: glial-derived neurotrophic factor; hADSCs: human adipose derived stem cells; hAFMSCs: amniotic fluid-derived mesenchymal stem cells; IB4: isolectin B4; IL: interleukin; NF-κB: nuclear factor kappa-light-chain-enhancer of activated B-cell transcription factor; NGF: nerve growth factor; pERK½: phosphorylated extracellular-regulated kinase ½; PGP9.5: protein gene product 9.5; p-p38-MAPK protein MAPK: mitogen-activated protein kinase; S100-CBP: S100 calcium-binding protein; TNF: tumor necrosis factor; SCI: spinal cord injury.

**Table 1 cells-11-03476-t001:** Summary of the reviewed neurological diseases and the types of cells used in experimental studies and/or clinical trials.

Disease	Cell Type or Tissue Used for Transplantation	Type of Research (Experimental vs. Clinical Trials)	Advantages	Disadvantages
AD	BM-MSCs HUC-MSCs iPSCs ESCs	Mostly experimental trials in mouse models and transgenic animals Clinical trials using HUC-MSCs	Decreased amyloid β plaques Decreased Tau phosphorylation Decreased tangles Increased microvesicles and extracellular vesicles Increased anti-inflammatory cytokines	Limited clinical trials
PD	Fetal Ventral mesencephalic grafts (fVM grafts) hiPSCs hESCs MSCs NSCs	Clinical trials by direct insertion of fetal tissues or adrenakl medullary cells Animal models for stem cells	Fetal grafts Successful in ameliorating the disease symptoms MSCs improved motor function in rats iPSCs transformed to dopaminergic neurons in monkey models improved the symptoms Adrenal medulla transplantation improved tremors and rigidity	fVM grafts: need large number of aborted fetuses (ethical issues) limited viability require immunosuppression produce dyskinesis iPSCs and ESCs produced poor results in animal models Adrenal medulla transplantation led to mental disorders
HD	Embryonic & fetal tissue grafts NSCs hESCs iPSCs + gene therapy	All experimental animal models	iPSCs in mice produced NCs for 6 months	Ethical issues and limited availability ESCs did not differentiate Need gene therapy and stem cell therapy
ALS	ADMSCs NPCs iPSCs Wharton’s Jelly MSCs	Animal models Clinical trials using Wharton’s Jelly MSCs	Delayed degeneration Production of neurotropic factors, spinal cord integration Female patients showed better clinical improvement	Mostly experimental animals, limited clinical trials
MS	aHSCs BM-MSCs NSCs hESCs iPSCs	EAE mice models and clinical trials	Decreased symptoms and immune infiltration Differentiation into oligodendrocytes Human trials were successful in young patients with active inflammation, low EDSS scores and no comorbidities	Need large scale clinical trials using aHSCs
TLE	ESCs Fetal GABAergic tissue Fetal neuronal cells Fetal medial ganglionic cells BM-MSCs	Experimental mice models	Production of GABA and mature neurons neurotransmitters, reduced apoptotic and inflammatory markers	Limited studies, no clinical trials
NP	BM-MSCs ADMSCs	Animal models	IV and intrathecal injection decreased glutamate excitotoxicity and microglial activity	Limited number of trials
BIS	NSCs	Clinical trial	Improvement	Limited number of studies

**Table 2 cells-11-03476-t002:** A summary of recent studies reported the paracrine effect of stem cells secretomes, exosomes, microvesicles, and extracellular vesicles on neurological disorders.

Secretome	Paracrine Effect	Techniques	References
Mesenchymal stem cell (MSC)-secreted factors the tissue inhibitor of metalloproteinase type 1 (TIMP-1)	Stimulate oligodendrogenesis from cultured primary adult neural stem cells (aNSCs) and oligodendroglial precursor cells (OPCs).	The MSC-conditioned medium	[240]
Mesenchymal stem cell (MSC)-secreted factors hepatocyte growth factor (HGF)	Stimulates the growth of oligodendrocytes and neurons	Conditioned medium from human MSCs (MSC-CM)	[241]
Adipose tissue-derived mesenchymal stem cells (ASC-CCM) secreted factors TNFα-stimulated gene-6 (TSG-6)	Neurovascular anti-inflammatory factor	Conditioned medium	[242]
Wharton’s jelly of the human umbilical cord (WJ-MSC) secreted factors	Induces neuronal maturation of SH-SY5Y neuroblastoma cells	Explant culture method	[243]
Bone marrow mesenchymal stem cells (BMSCs) secreted factors	Induced protection for neurons against oxygen-glucose deprivation (OGD) through in part promoting secretion of VEGF	Hypoxia conditioning medium	[244]
iPSC-derived neuronal models of Alzheimer’s disease secretomes Aβ peptides and extracellular tau.	Induces synaptic dysfunction	In vivo injection into the rat brain	[245]
MSC secreted exosomes and microvesicles (miR-133b)	Enhances neural functional recovery through increase nitrite remodeling, neurogenesis, and angiogenesis	In vivo injection in rats	[246]
MSC-extracellular vesicles (PD-L1, galectin-1, and membrane-bound TGF-beta)	Activates T-immune cell and induce peripheral tolerance to prevent tissue damage, therefore, it modulates the immune response in encephalomyelitis	Conditioned medium	[247]
Human adipose MSC-extracellular vesicles (neprilysin)	The active form of neprilysin acts to decrease β-amyloid peptide accumulation in the brain	In vitro model of Alzheimer’s disease	[37]
Rat BM MSC- extracellular vesicles(Catalase)	The catalase acts as an antioxidant protects neuron from the damaging effect induced by β-amyloid	Co-cultures of rat hippocampal neurons and MSCs	[248]

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
