# Peer review of "Stem-Cell-Based Therapy: The Celestial Weapon against Neurological Disorders"

_cells, 2022, doi:10.3390/cells11213476_

Round 1
Reviewer 1 Report
This review manuscript by Zayed et al. comprehensively summarized the current research on the use of stem cells both experimentally and in clinical trials for the treatment of selected neurological disorders. Specifically, the authors described the need to try and tailor different types of cells to repair the specific defect characteristic of each neurological disease, and discussed in details on the cell sources, outcomes, delivery approaches, among others. They further summarized the current pitfalls for transplantation, as well as effective methods to improve the cell survival, which is critical for therapeutic benefits.
Overall, the manuscript is well-structured and well-written, and has systematically described the studies on the transplanted selected cells for treating various neurological diseases. However, there are still some minor concerns that may dampen the integrity of this review manuscript:
- About Figures: the authors made their Figures with respect to stem cell and MS. I do believe that the rationale or an overall panorama of stem cell transplantation in treating neurological diseases shall be depicted as individual figures. Also, introductions on other neurodegenerative diseases such as AD, PD, HD and ALS would be better described if any figure provided.
- please pay attention to the usage of abbreviations, for example:
Neuronal Stem Cells (NSCs); Human Embryonic Stem Cells (hESCs); Induced Pluripotent Stem Cells (iPSC) have been mentioned in a lot of places throughout the manuscript. The abbreviations can be mentioned at the first place where they occur, and later directly used in the following part of the manuscript.
Others: the usage of abbreviations for parkinson’s disease (for example: line 290) and others shall be noticed.
- proofreading is required with particular attention to grammer, sentence structure and hypos.
For example: line 277: a PD mice models
And etc.
Reviewer 2 Report
It is a very interesting and well-written manuscript, covering very thoroughly the potential of stem-cell based therapy against neurodegenerative diseases.
The paper is comprehensive, up-to-date, and well-presented, with little or no serious problems. Although there are published paper on this topic; I must say this review not only covers up-to date clinical trials, but also it closely represented the recent trends in stem-cell research. The following suggestions can be made to considerably improve this situation:
1. I've noticed a few sticky words, which I feel are the result of formatting problems.
I recommend that the writers remove these.
2. Authors are advised to double-check citations throughout the text and must be updated with recent references.
3. The resolution of the images could be increased by using good quality software.
4. The text in the figure legend could be expanded so that it can make the figures more understandable.
